# Vision-DeepResearch: Incentivizing DeepResearch Capability in Multimodal Large Language Models

**Wenxuan Huang** [* † 1 2] **Yu Zeng** [* 3] **Qiuchen Wang** [* 3] **Zhen Fang** [3] **Shaosheng Cao** [✉ 4 5] **Zheng Chu** [6]
**Qingyu Yin** [7] **Shuang Chen** [8] **Zhenfei Yin** [9] **Lin Chen** [3] **Zehui Chen** [3] **Yao Hu** [4] **Philip Torr** [9] **Feng Zhao** [✉ 3]
**Wanli Ouyang** [✉ 1 10]

wxhuang0616@gmail.com (Wenxuan Huang)

*: Equal Contribution    †: Project Leader    ✉: Corresponding Author

## Abstract

Multimodal large language models (MLLMs) have achieved remarkable success across a broad range of vision tasks. However, constrained by the capacity of their internal world knowledge, prior work has proposed augmenting MLLMs by "reasoning-then-tool-call" for visual and textual search engines to obtain substantial gains on tasks requiring extensive factual information. However, these approaches typically define multimodal search in a naive setting, assuming that a single full-level or entity-level image query and few text query suffices to retrieve the key evidence needed to answer the question, which is unrealistic in real-world scenarios with substantial visual noise. Moreover, they are often limited in the reasoning depth and search breadth, making it difficult to solve complex questions that require aggregating evidence from diverse sources. Building on this, we propose *Vision-DeepResearch*, which proposes one **new multimodal deep-research paradigm**, *i.e.*, performs multi-turn, multi-entity and multi-scale visual and textual search to robustly hit real-world search engines under heavy noise. Our Vision-DeepResearch supports dozens of reasoning steps and hundreds of engine interactions, while internalizing deep-research capabilities into the MLLM via cold-start supervision and RL training. It substantially outperforming existing multimodal deep-research MLLMs, and workflows built on strong closed-source foundation model such as GPT-5, Gemini-2.5-pro and Claude-4-Sonnet. The code will be released in https://github.com/Osilly/Vision-DeepResearch.

## 1. Introduction

Multimodal large language models (MLLMs) have achieved substantial success on a wide range of real-world tasks (Liu et al., 2023; Huang et al., 2025; Bai et al., 2025). However, due to their limited internal world knowledge, complex fact-intensive VQA remains a major challenge (Jiang et al.; Wu et al., 2025a; Geng et al., 2025; Narayan et al., 2025). Recent multimodal deep-research MLLM works (Wu et al., 2025a; Geng et al., 2025; Narayan et al., 2025) have proposed equipping MLLMs with the "reasoning-then-tool-call" paradigm (*i.e.*, ReAct (Yao et al., 2022)), where models use external tools as one action after reasoning to query search engines and obtain factual observations. This substantially improves MLLMs' performance on VQA problems that require extensive real-world knowledge.

However, existing works face two key issues. First, *they formulate multimodal search under an overly simple setting*. Image queries are treated as full-image (or entirety-level) retrieval, and a small number of text queries is assumed to suffice. This overlooks a critical challenge in realistic, noisy search engines: the **hit-rate problem**. This issue is evident in practice. As illustrated in Fig. 1 (A.1), full-image retrieval can be dominated by irrelevant visual noise. In real user scenarios, it is unrealistic to assume that the entire image content is aligned with the target information. Furthermore, search engines exhibit substantial hit-rate variability. Even when querying the same visual or textual entity, retrieval

[1]CUHK MMLab, Hong Kong, China [2]East China Normal University, Shanghai, China [3]MoE Key Lab of BIPC, University of Science and Technology of China, Hefei, China [4]Xiaohongshu Inc., Shanghai, China [5]Tsinghua University, Beijing, China [6]Harbin Institute of Technology, Harbin, China [7]Zhejiang University, Hangzhou, China [8]University of California, Los Angeles, CA, USA [9]University of Oxford, Oxford, UK [10]Shenzhen Loop Area Institute, Shenzhen, China. Correspondence to: Shaosheng Cao <caoshaosheng@xiaohongshu.com>, Feng Zhao <fzhao956@ustc.edu.cn>, Wanli Ouyang <wlouyang@ie.cuhk.edu.hk>.

*Proceedings of the 43rd International Conference on Machine Learning*, Seoul, South Korea. PMLR 306, 2026. Copyright 2026 by the author(s).

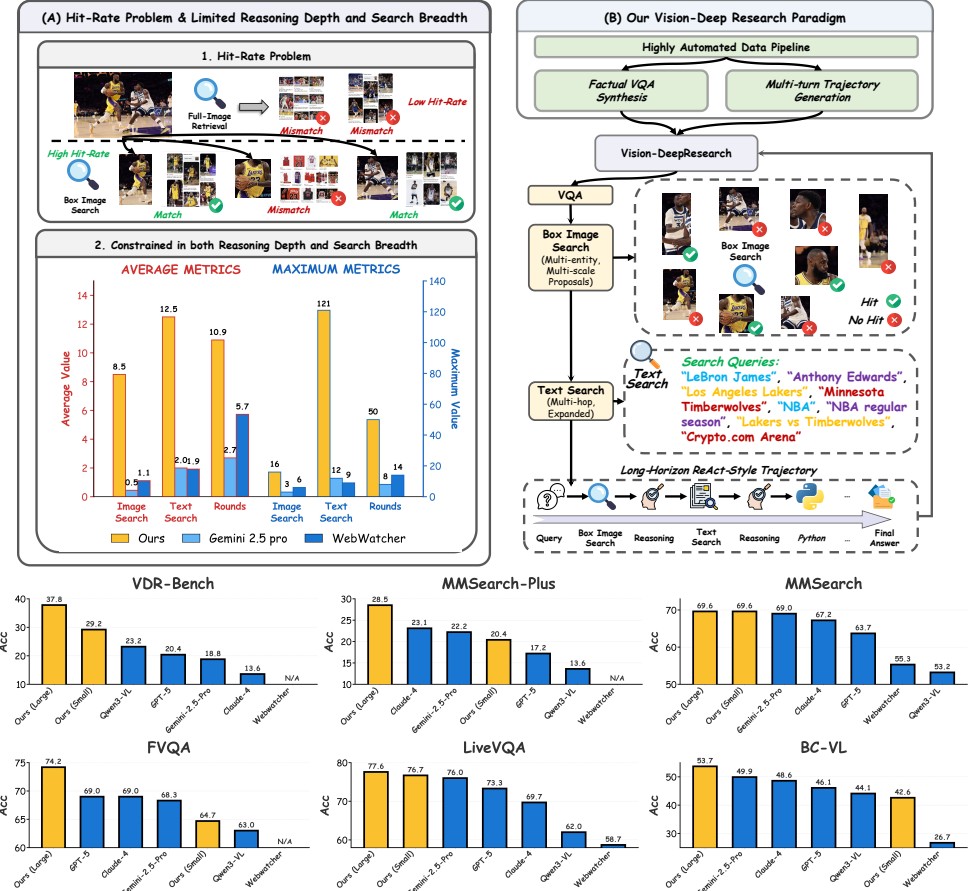

*Figure 1.* **Panel A**: We identify two key limitations of existing multimodal deep-research paradigms for image search. First, prior multimodal deep-research MLLMs largely ignore the search engine hit-rate problem. In image retrieval, a single full-image or even entity-level query often fails to retrieve the required evidence; moreover, querying different-scale crops of the same entity can yield highly variable results. Second, existing methods are constrained in both reasoning depth and retrieval breadth, typically producing only short trajectories. In contrast, our approach supports dozens of reasoning steps and hundreds of engine interactions, leading to substantially stronger performance. **Panel B**: Pipeline Overview. We synthesize high-quality VQA instances and multi-turn trajectories, and then integrate multimodal deep-research capabilities into an MLLM via SFT and RL training. This enables long-horizon reasoning that performs multi-turn, multi-entity, and multi-scale visual and textual search. **Bottom Image**: Performance Comparison. Our model achieves the SoTA performance on six benchmarks with a comparatively smaller parameter. The our "Large" and "Small" models correspond to the 30B-A3B and 8B parameter scales, respectively, while "Qwen3-VL" and "WebWatcher" refer to Qwen3-VL-30B-A3B-Thinking and WebWatcher-32B, respectively. ***All models are evaluated fairly under the same agentic-reasoning setting***.

results can differ markedly across query scales, and obtaining the required evidence from real-world engines is often non-trivial.

Second, *existing methods are constrained in both reasoning depth and search breadth, making it difficult to perform complex multi-hop deep-research and to aggregate evidence from multiple information sources.* Rather than treating retrieval as a one-off operation, it should be modeled as a trial-and-error process that adaptively explores multi-scale search and iteratively refines queries based on intermediate results, enabling it to better handle noisy, unstable search environments and ambiguous inputs. As shown in Fig. 1 (A.2), our proposed Vision-DeepResearch substantially increases both the number of reasoning steps and the number of engine interactions compared to prior multimodal deep-

research MLLMs and agentic workflows. This enables one 30B-A3B–scale and even a 8B-scale model to achieve State-of-The-Art (SoTA) performance across multiple multimodal factual benchmarks (see Fig. 1 (bottom)).

To address the mentioned issues, *we design one new multimodal deep-research paradigm.* As presented in Fig. 1 (B), we design the highly automated factual VQA synthesis and multi-turn trajectory generation pipelines to obtain high-quality training data, and then integrate multi-turn, multi-entity, and multi-scale visual and textual search capabilities into a base MLLM through cold-start supervision and reinforcement learning (RL) training, yielding a strong multimodal deep-research MLLM, **Vision-DeepResearch**. We adopt an entity-level, stringent image verification and filtering pipeline for candidate question synthesis. To expand

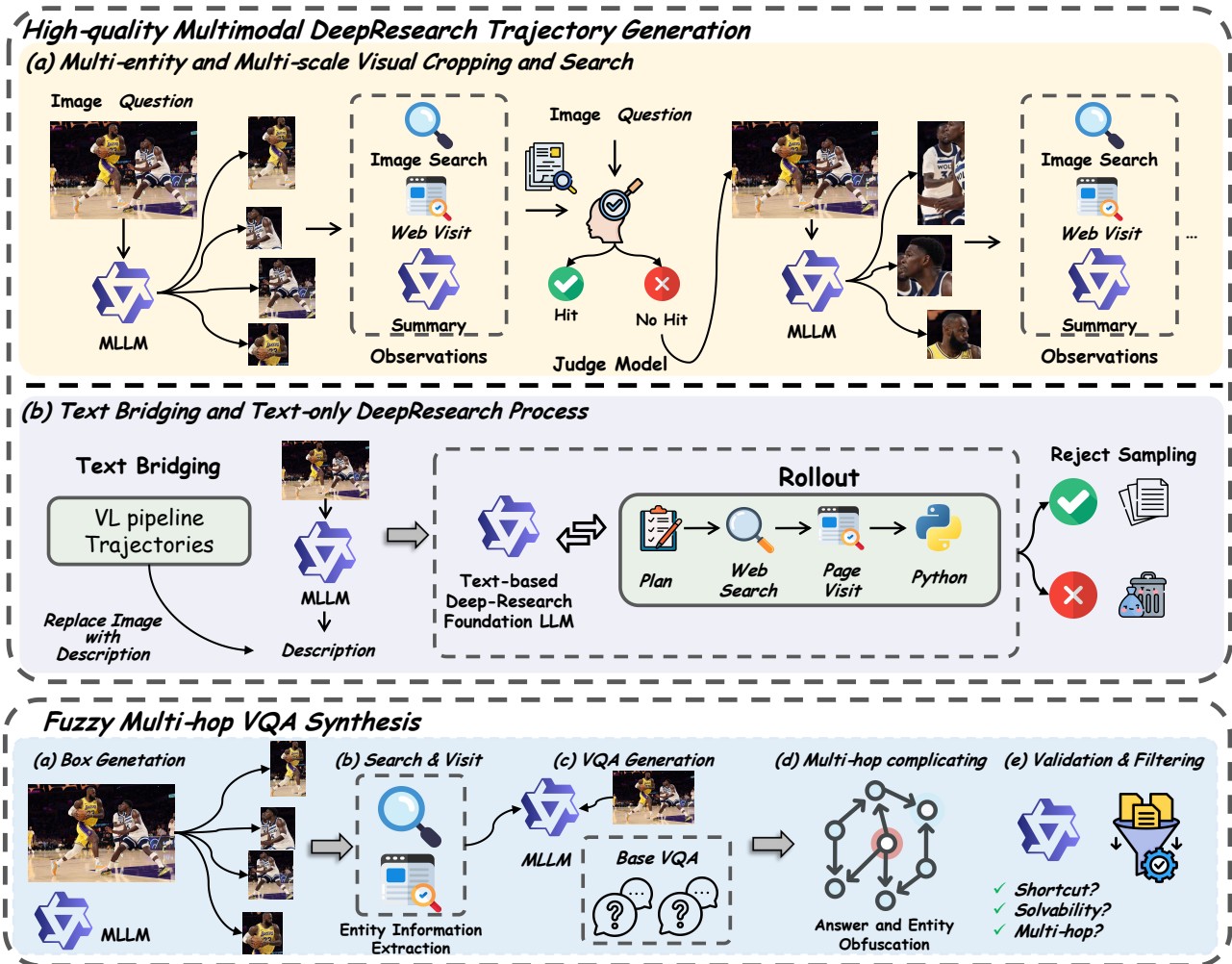

Figure 2. Our Data Pipeline. As shown in the top panel, we construct a complete multimodal deep-research synthesis pipeline. Leveraging the capabilities of an MLLM and a text-based DeepResearch foundation LLM, we generate long-horizon, multi-tool trajectories. As shown in the bottom panel, we obtain high-quality factual VQA instances via a rigorous verification and obfuscation procedure, which are then used for trajectory synthesis and RL training.

the multi-hop structure of the textual component, we further perform random walks over real search engines and real web pages, together with joint entity and answer obfuscation, to obtain high-quality factual VQA data. Moreover, we induce existing MLLMs to generate multi-entity, multi-scale visual region proposals to efficiently probe visual search engines and collect visual retrieval trajectories. We also introduce an obfuscated termination strategy to control the depth of visual retrieval. We then bridge modalities by leveraging a strong deep-research LLM foundation model to produce the corresponding text-search trajectories, ultimately yielding complete multimodal deep-research cold-start trajectories.

Extensive experiments show that our proposed Vision-DeepResearch significantly outperforms all prior multimodal deep-research MLLMs on six factual benchmarks, and even surpasses agent workflows built on strong closed-source models such as GPT-5, Gemini-2.5-Pro, and Claude-4-Sonnet. *We believe our work can inspire the research community deeply.*

## 2. Method

In this section, we present the pipeline for constructing our vision deep research agent, which is capable of performing long-horizon vision–language reasoning in realistic and noisy web environments. We first discuss the design motivation (Sec. 2.1), then describe the data generation process in detail (Sec. 2.2), and finally outline the training strategies (Sec. 2.3).

### 2.1. Motivation

Realistic multimodal deep-research systems require gather information under noisy web conditions. However, as shwon in Sec. 1, existing works perform search tools to short-horizon or single-shot operations, leading to brittle retrieval

behavior and premature convergence in complex multimodal reasoning tasks. We highlight two key issues below.

**Coarse search strategy**. In real-world web environments, search environment is inherently noisy and unstable. Images often contain multiple visual entities with cluttered backgrounds and large scale variations, while search engines may return inconsistent results even for visually similar queries, making reliable retrieval particularly challenging. Humans naturally cope with such conditions through an iterative and exploratory search process: when initial attempts fail, they progressively refine their queries by cropping different regions, adjusting scales, or focusing on alternative visual cues until sufficient evidence is accumulated to identify the target entity. The same holds for textual search: reaching the desired target via a search engine often requires multiple attempts. Even minor word-level modifications to a query can lead to substantially different retrieved content.

A robust multimodal deep-research system can obtain substantial benefits by mirroring this human search behavior. Instead of treating retrieval as a one-shot operation, it should model retrieval as a trial-and-error process, adaptively exploring multi-scale regions and refining queries based on intermediate results to better handle noisy, unstable search environments and ambiguous inputs. Unfortunately, in visual search side, this problem is particularly even worse. Most prior work adopts a single-pass, full-image or entity retrieval paradigm, where the model issues a single visual query to the search engine using the original image. This approach is highly fragile in real-world web environments containing multiple visual entities.

**Lack of optimization for long-horizon engine interaction**. Most existing training dataset synthesis in multimodal deep-research MLLMs (Jiang et al.; Wu et al., 2025a; Geng et al., 2025; Narayan et al., 2025) contain search trajectories that are limited to short contexts or an average of fewer than five retrieval rounds. As a result, models trained on such data tend to prematurely terminate exploration in complex tasks, settling for partially relevant evidence rather than systematically exploring alternative visual or textual queries. However, there currently exists no systematic framework for synthesizing trajectories that involve dozens of reasoning steps and both visual and textual search engine interactions, which are essential for authentic deep research scenarios.

In contrast, deep-research LLMs (Team et al., 2025) specialized for text-based deep research have demonstrated strong ReAct-style capabilities, enabling them to iteratively plan, search, and reflect over long contexts, often involving tens of tool invocations. Nevertheless, current MLLMs struggle to transfer such long-horizon reasoning behavior from text-only tasks to multimodal settings.

To address these challenges, we design a new data and

training paradigm consisting of the following components:

- **Multi-entity and multi-scale visual cropping and search**. We enable robust trial-and-error visual retrieval in noisy environments by performing visual search over multiple scales and regions, thereby increasing the hit rate of target visual entities.

- **Long-horizon trajectory construction**. We leverage the strong localization capabilities of MLLMs together with the deep retrieval behaviors of text-based deep research models, and seamlessly transfer their long-horizon ReAct-style reasoning to the visual domain via image-description-based context window sharing.

### 2.2. Data Pipeline

As shown in Fig 2, our data pipeline constructs *long-horizon multimodal deep-research trajectories* by combining visual search with text-based deep-research reasoning, bridged through image descriptions.

#### 2.2.1. HIGH-QUALITY MULTIMODAL DEEPRESEARCH TRAJECTORY GENERATION

**Multi-entity and Multi-scale Visual Cropping and Search**. Given an input image $I$, the question $q$ and the prompt $p_v$ to induce one MLLM to generate single-turn ReAct-style context for visual toll call, we first generate one reasoning $R$ and then localize regions relevant to the query and generate multiple bounding boxes $S_b = \{I_b^1, \ldots, I_b^n\}$, including fine-grained entity-level regions $I_b$. Each cropped regions set in $t$-th step defines a visual action $A^t = \text{Tool-Call}(S_b^t)$, which is submitted to the visual search tool pipeline, where $t_v \in \{1, \ldots, T_v\}$ and $T_v$ is the final step of the visual tool pipeline.

We denote the observation returned by the visual tool pipeline Vision-Pipeline at step $t$ as:

$$\mathcal{O}^{t_v} = \text{Vision-Pipeline}(A^{t_v}), \qquad (1)$$

and the cumulative visual evidence up to step $t_v$ as:

$$\mathcal{V}^{t_v} = \{\mathcal{O}^1, \ldots, \mathcal{O}^{t_v}\}, \qquad (2)$$

where the tool list include three sequential execution tools, *i.e.*, the visual search tool, website visit tool and website summary tool. The visual search tool takes a cropped image region as input and returns the matched webpage URL, we then use a website visit tool to fetch the page content in markdown format. The returned content is typically long and cluttered with image links, and passing it directly to the MLLM can easily exceed the context window. To mitigate this, we use an auxiliary MLLM to summarize the webpage and verify the correspondence between the cropped image query and the matched images on the page, so as to extract

the most relevant evidence while filtering out irrelevant content.

When we finish the single-turn ReACT-style context, we use the induction prompt $p_v$ again to generate the next-turn context. Finally, the visual tool trajectory is represented as:

$$\mathcal{C}_{\text{vision}} = \{I, q, p_v, R^1, A^1, \mathcal{O}^1, \dots, p_v, R^{T_v}, A^{T_v}, \mathcal{O}^{T_v}\}, \quad (3)$$

To control visual search depth, an external judge model evaluates whether the accumulated evidence $\mathcal{V}_t$ is sufficient to support downstream text-based reasoning steps. Conditioned on the original image $I$, the question $q$, the ground truth $a_{\text{true}}$ and the collected evidence, the judge outputs a binary *hit signal*.

$$h^{t_v} = \text{Judge}(I, q, \mathcal{V}^{t_v}, a_{\text{true}}) \in \{0, 1\}, \quad (4)$$

In the Judge stage, we ask the model to make a relatively lenient determination of whether the currently available visual evidence contains sufficient information to answer the question. If $h^{t_v} = 0$, the pipeline continues with additional visual search actions $A^{t_v+1}$, while if $h^{t_v} = 1$, the final step of the visual tool pipeline $T_v$ is set to current $t_v$ and the visual tool process terminates. With this strategy, we aim to retrieve sufficiently informative evidence from the visual search engine to support the subsequent text-based reasoning process.

**Text Bridging and Text-only DeepResearch Process**. To leverage the strong ReAct-style capability of the text-based deep-research foundation LLM, we bridge the above visual trajectory $\mathcal{C}_{\text{vision}}$ to the text-only context. We first generate a detailed textual description $D$ for the input image $I$, while replacing $I$ with $D$ and removing the induction prompts $p$ which in $\mathcal{C}_{\text{vision}}$, keep the rest of the trajectory unchanged as the bridged context, *i.e.*, reasoning $R^{t_v}$, actions $A^{t_v}$ and observations $\mathcal{O}^{t_v}$ remain the same. We then sent the bridged context and the corresponding prompt $p_t$ to induce the text-based deep-research foundation LLM yields the subsequent text-based trajectory. The textual tool trajectory of text-based deep-research foundation LLM can denote as:

$$\begin{aligned}
\mathcal{C}_{\text{text}} &= \\
&\{D, q, R^1, A^1, \mathcal{O}^1, \dots, R^{T_v}, A^{T_v}, \mathcal{O}^{T_v}, \\
&p_t, R^{T_v+1}, A^{T_v+1}, \mathcal{O}^{T_v+1}, \dots, R^{T_v+T_t}, A^{T_v+T_t}, a_{\text{output}}\},
\end{aligned} \quad (5)$$

where $T_t$ is the number of steps generated by the foundation LLM and $a_{\text{output}}$ is the model's final answer output. The textual tool list involving web search, website visit&summary and python code.

Finally, we merge the trajectories $\mathcal{C}_{\text{vision}}$ and $\mathcal{C}_{\text{text}}$ to obtain the full multimodal deep-research trajectory:

$$\begin{aligned}
\mathcal{C}_{\text{multimodal}} &= \\
&\{I, q, R^1, A^1, \mathcal{O}^1, \dots, R^{T_v}, A^{T_v}, \mathcal{O}^{T_v}, \\
&R^{T_v+1}, A^{T_v+1}, \mathcal{O}^{T_v+1}, \dots, R^{T_v+T_t}, A^{T_v+T_t}, a_{\text{output}}\},
\end{aligned} \quad (6)$$

We then apply rejection sampling to select trajectories $\mathcal{C}_{\text{multimodal}}$ for cold-start training, *i.e.*, an LLM verifies whether the final trajectory output $a_{\text{output}}$ matches the ground-truth answer $a_{\text{true}}$. Consistent trajectories are retained in the training set, while inconsistent ones are discarded. Furthermore, we also incorporate text-only deep-research trajectories generated by the original question $q$ into the foundation LLM.

### 2.2.2. VERIFIED FACTUAL VQA GENERATION

**Factual VQA Verification**. First, we curate images from multiple open-source datasets (Hu et al., 2023; Marino et al., 2019; Fu et al., 2026; Wu et al., 2025a), focusing on real-world, high-quality, complex images with multiple entities. We filter out images smaller than $224 \times 224$, and then use an MLLM as a selector, guided by predefined criteria, to identify high-quality real-world images and remove overly trivial cases. We then further filter the candidates along two dimensions. We directly feed the VQA instance to an MLLM. If it can answer correctly without external evidence, we discard the sample. Moreover, we submit the full image to an image search engine. If the retrieved results perfectly match the full-image query, we also discard the sample.

Applying these criteria yields a higher-quality factual VQA dataset. We use a subset of the resulting samples for trajectory synthesis (Sec. 2.2.1) and RL training (Sec. 2.3.2), while using the remaining images (discarding their original QAs) for Fuzzy Multi-hop VQA Synthesis.

**Fuzzy Multi-hop VQA Synthesis**. For each retained image, we first prompt an MLLM to propose a set of entity-level candidate bounding boxes. We then crop these regions at multiple scales and perform image search. We match the retrieved images against the corresponding crops. If they refer to the same entity, we retain the box and its associated entity. We then use an MLLM to generate a simple, unambiguous entity-level question, *e.g.*, 'What is the name of the cat in the image?".

The entity-level questions obtained above are often overly explicit and simplistic, deviating from real user queries. We therefore further obfuscate both the entity and the question in two ways.

1. *Answer obfuscation*, which increases the required reasoning depth by chaining relations around the answer (*e.g.*, "What is the name of the teacher of the cat owner's daughter?").

2. *Entity obfuscation*, a technique commonly used in text-only deep-research LLMs (Wu et al., 2025c;b; Li et al., 2025; Tao et al., 2025b), where we perform random walks over webpages to replace the original entity with related entities along multi-hop links (*e.g.*, "The cat's owner works at A, and the owner's daughter studies at B. So what is the

*Table 1.* Benchmark results across different Settings with improvement (Δ, compared with base MLLM in agentic workflow setting). The best results are highlighted in **bold**, and the second-best results are underlined.

| Model | VDR | FVQA | MMSearch+ | MMSearch | LiveVQA | BC-VL | Avg. |
|---|---|---|---|---|---|---|---|
| **Direct Answer** | | | | | | | |
| GPT-5 | 9.8 | 57.3 | 19.1 | 33.3 | 57.5 | 47.2 | 37.4 |
| Gemini-2.5 Pro | 8.0 | 60.7 | 14.5 | 39.8 | 60.3 | 43.1 | 37.7 |
| Gemini-2.5 Flash | 6.2 | 47.7 | 8.1 | 30.4 | 51.0 | 37.1 | 30.1 |
| Claude-4-Sonnet | 2.0 | 35.3 | 4.0 | 18.7 | 38.5 | 29.3 | 21.3 |
| Claude-3.7-Sonnet | 4.6 | 36.7 | 4.0 | 21.1 | 38.0 | 32.3 | 22.8 |
| Qwen3-VL-8B-Instruct | 2.8 | 28.0 | 3.2 | 15.2 | 41.0 | 25.1 | 19.2 |
| Qwen3-VL-8B-Thinking | 5.6 | 24.0 | 2.7 | 15.8 | 43.3 | 25.1 | 19.4 |
| Qwen3-VL-30B-A3B-Instruct | 3.8 | 34.7 | 3.2 | 18.7 | 42.7 | 29.6 | 22.1 |
| Qwen3-VL-30B-A3B-Thinking | 4.4 | 32.7 | 4.5 | 19.3 | 49.0 | 34.6 | 24.1 |
| **RAG Workflow** | | | | | | | |
| Gemini-2.5-flash | – | – | – | 43.9 | 41.3 | 12.1 | – |
| Claude-3.7-Sonnet | – | – | – | 32.7 | 30.3 | 10.0 | – |
| Qwen-2.5-VL-72B | – | – | – | 29.2 | 35.7 | 10.2 | – |
| **Agent Workflow** | | | | | | | |
| GPT-5 | 20.4 | 69.0 | 17.2 | 63.7 | 73.3 | 46.1 | 48.3 |
| Gemini-2.5 Pro | 18.8 | 68.3 | 22.2 | 69.0 | 76.0 | 49.9 | 50.7 |
| Gemini-2.5 Flash | 16.3 | 68.0 | 19.9 | 64.0 | 73.0 | 44.6 | 47.6 |
| Claude-4-Sonnet | 13.6 | 69.0 | 23.1 | 67.2 | 69.7 | 48.6 | 48.5 |
| Claude-3.7-Sonnet | 27.2 | 67.3 | 17.2 | 63.7 | 72.0 | 50.4 | 49.6 |
| Qwen3-VL-8B-Thinking | 17.6 | 51.3 | 12.2 | 45.6 | 56.3 | 37.1 | 36.7 |
| Qwen3-VL-30B-A3B-Thinking | 23.2 | 63.0 | 13.6 | 53.2 | 62.0 | 44.1 | 43.2 |
| **Multimodal DeepResearch MLLM** | | | | | | | |
| MMSearch-R1-7B | – | 58.4 | – | 53.8 | 48.4 | – | – |
| Webwatcher-7B | – | – | – | 49.1 | 51.2 | 20.3 | – |
| Webwatcher-32B | – | – | – | 55.3 | 58.7 | 26.7 | – |
| **Ours** | | | | | | | |
| Qwen3-VL-8B-Instruct (Agentic) | 17.0 | 58.7 | 11.3 | 52.0 | 63.0 | 38.6 | 40.1 |
| Vision-DeepResearch-8B (Ours) | 29.2 | 64.7 | 20.4 | **69.6** | 76.7 | 42.6 | 50.5 |
| Δ | **+12.2** | **+6.0** | **+9.1** | **+17.6** | **+13.7** | **+4.0** | **+10.4** |
| Qwen3-VL-30B-A3B-Instruct (Agentic) | 20.2 | 57.7 | 10.0 | 55.0 | 60.0 | 42.6 | 40.9 |
| Vision-DeepResearch-30B-A3B (Ours) | **37.8** | **74.2** | **28.5** | **69.6** | **77.6** | **53.7** | **56.9** |
| Δ | **+17.6** | **+16.5** | **+18.5** | **+14.6** | **+17.6** | **+11.1** | **+16.0** |

cat's name?").

However, repeatedly increasing complexity purely via answer chaining can lead to rigid, templated reasoning patterns. In contrast, entity obfuscation alone can introduce cross-source shortcuts, where the final answer can be inferred via consistency checks over multiple textual entities, without requiring any visual evidence. We therefore adopt an interleaved obfuscation strategy that alternates between answer and entity obfuscation.

By combining both answer obfuscation and entity obfuscation, the resulting questions become significantly more aligned with complex, real-world inquiry patterns. For instance, a simple root question like "What brand is the football in the picture?" can be transformed through iterative answer obfuscation into "What is the name of the founder of the football brand in the picture?" and further to "What did the founder of the football brand... do in Wilmslow, Cheshire, England?" Applying entity obfuscation ultimately yields a highly complex query: "There is a town in Cheshire,

England, located 11 miles south of Manchester. What is the famous thing the founder of the football brand in the picture did in this town?" To generate such intricate questions, we look to real-world question design, where humans typically (1) decide the assessment target, (2) search to verify answerability, (3) draft multiple candidate questions to select the best one, and (4) attempt to solve it themselves. We emulate this process via an automated pipeline: an MLLM first extracts retrieval keywords from the current entity and queries external sources to collect additional information. Subsequently, another MLLM proposes multiple candidate questions conditioned on the retrieved evidence, and a judge MLLM selects the most reasonable and objective one. We iterate this pipeline during interleaved answer and entity obfuscation to produce the final complex question-answer pairs. Finally, we merge the original image with these pairs to construct fuzzy multi-hop VQA problems. These resulting instances are treated as higher-quality samples, serving critically in both multimodal deep-research trajectory synthesis and Reinforcement Learning (RL) training.

*Table 2.* Ablation study on rollout pipeline.

| Setting | VDR | MMS+ | BC-VL | Avg. |
|---|---|---|---|---|
| Direct Answer | 4.8 | 3.6 | 27.6 | 12.0 |
| Image Caption + TS | 9.0 | 16.7 | 42.0 | 22.6 |
| Image + TS | 11.6 | 15.8 | 44.6 | 24.0 |
| WIS | 11.8 | 10.0 | 26.1 | 16.0 |
| WIS+TS | 16.0 | 23.5 | 48.4 | 29.3 |
| CIS | 15.4 | 22.7 | 30.8 | 23.0 |
| CIS+TS (Ours) | **37.8** | **28.5** | **53.7** | **40.0** |

*Table 3.* Ablation study on rollout pipeline across additional benchmarks.

| Setting | FVQA | MMSearch | LiveVQA | Avg. |
|---|---|---|---|---|
| Direct Answer | 30.7 | 15.8 | 42.0 | 29.5 |
| WIS | 44.3 | 27.5 | 47.3 | 39.7 |
| WIS+TS | 67.2 | 61.4 | 71.3 | 66.6 |
| CIS | 61.8 | 40.4 | 61.7 | 54.6 |
| CIS+TS | **74.2** | **69.6** | **77.6** | **73.8** |

## 2.3. Training

We use the generated multimodal deep-research trajectories and verified VQA to train our Vision-DeepResearch model using a combination of supervised fine-tuning (SFT) and reinforcement learning (RL).

### 2.3.1. SUPERVISED FINE-TUNING

We first perform SFT to teach the model the fundamental behaviors required of a multimodal deep-research MLLM. Following the pipeline described in Sec. 2.2, we collect 30K high-quality trajectories. Each trajectory $\mathcal{C}_{\text{multimodal}}$ consists of: a question, an initial image and the multi-turn deep-research steps (including both multimodal and text-only trajectories). For data mixing, we sample verified fact-centric VQA problems from existing VQA datasets and augment them with multimodal deep-research trajectories, yielding 16K instances. We also sample text-only QA problems and generate corresponding text-only trajectories, resulting in 8K instances. Finally, we sample 6K fuzzy VQA instances and similarly populate them with trajectories. The VQA filter process and fuzzy VQA synthesis described in Sec. 2.2.2.

The model is trained by minimizing the standard autoregressive cross-entropy loss (CE loss). The goal of SFT is to guide the model to learn multi-turn, multi-entity and multi-scale patterns, integrate visual and textual evidence during reasoning, and develop long-horizon planning behaviors for visual tasks that are analogous to those exhibited by text-based deep research models.

### 2.3.2. REINFORCEMENT LEARNING

**High-throughput Asynchronous Rollout Architecture Design**. Multi-turn agentic-reasoning RL is typically bottlenecked by rollouts, *i.e.*, long reasoning horizons and fre-

quent tool calls introduce substantial latency, and naive synchronous rollouts can severely stall the event loop. To this end, we design a high-throughput multi-threaded asynchronous rollout pipeline building on rLLM framework (Tan et al., 2025). It dispatches tasks via a queued scheduler and maintains a tool pool to support concurrent multi-tool calls within a single action (*e.g.*, querying search results for multiple image crops in parallel), returning observations asynchronously. By offloading potentially blocking operations for event loop, our asynchronous design achieves over $10\times$ higher rollout throughput than synchronous rollouts, to achieve the efficient RL training of our Vision-DeepResearch.

**Training Recipe**. To further refine the agent's behavior, we apply RL training with Group Relative Policy Optimization (GRPO) (Shao et al., 2024; Guo et al., 2025) with Leave-One-Out trick (Ahmadian et al., 2024; Luo; Chen et al., 2025) on the post-SFT Vision-DeepResearch model. We perform reinforcement learning using 15K high-quality VQA instances, where 10K filtered from existing VQA dataset and 5K obtained from fuzzy VQA synthesis pipeline. During training, the model interacts with a real online search environment (including visual search, text search, and website visiting) and samples long-horizon rollout trajectories. We cap the maximum horizon, context length and single-turn response length at 50 turns, 64K and 4K tokens, respectively.

For reward design, we adopt LLMs-as-Judge paradigm, where a judge model determines whether the final answer matches the reference answer and provides the corresponding reward signal. During training, we use a pure accuracy reward, *i.e.*, the model receives a reward of 1.0 if the generated answer is correct, and 0.0 otherwise.

**Engineering Trick**. During RL training, we explore many tricks to keep stable.

*1. Trajectory Rollout Rnterrupted.* During RL training, we encounter a severe long-tail issue, where a small fraction of trajectories dominates the wall-clock time of an entire rollout batch. This long tail typically arises in two cases. First, repetitive text: the model may enter degenerate loops, repeatedly generating near-duplicate responses until hitting the context limit. We mitigate this via an n-gram–based repetition detector with a minimum character-length threshold; once repetition is detected and the response exceeds the threshold, we immediately terminate the trajectory. Second, cascading format/tool-call failures: the model may repeatedly produce invalid formats or incorrect tool invocations, persisting until the maximum turn or length budget is exhausted. For this case, we track consecutive errors and terminate the trajectory once the error count reaches three. For other cases (*e.g.*, isolated formatting mistakes or occasional tool-call errors), we return an explicit observation (*e.g.*, "format error, please try again") and allow the model

*Table 4.* Ablation results on training data and methods.

| Model / Setting | VDR | MMS+ | BC-VL | FVQA | MMSearch | LiveVQA | Avg. |
|---|---|---|---|---|---|---|---|
| Qwen3-VL-30B-A3B-Instruct | 20.2 | 10.0 | 42.6 | 57.7 | 55.0 | 60.0 | 40.9 |
| +16K VQA traj. (SFT) | 24.4 | 23.5 | 50.9 | 68.2 | 60.0 | 72.0 | 49.8 |
| +8K QA traj. (SFT) | 27.0 | 23.5 | 50.1 | 70.3 | 62.4 | 74.3 | 51.3 |
| +6K fuzzy VQA traj. (SFT) | 33.2 | 26.0 | 51.4 | 73.0 | 65.6 | 77.6 | 54.5 |
| +RL training | **37.8** | **28.5** | **53.7** | **74.2** | **69.6** | **77.6** | **56.9** |

*Table 5.* Performance comparison on document-based VQA benchmarks.

| Model | SlideVQA | InfoVQA | MML-D | Avg. |
|---|---|---|---|---|
| VDocRAG | 48.0 | 56.2 | 14.5 | 39.6 |
| ViDoRAG | 82.5 | 79.1 | 37.9 | 66.5 |
| VRAG-RL | 56.2 | - | 24.9 | - |
| Ours | **93.2** | **90.4** | **58.3** | **80.6** |

*Table 6.* Ablation study on maximum turns.

| Max Turns | VDR | MMS+ | BC-VL | FVQA | MMS | LiveVQA | Avg. |
|---|---|---|---|---|---|---|---|
| 1 | 1.0 | 1.0 | 12.0 | 22.5 | 6.8 | 16.0 | 9.9 |
| 10 | 14.2 | 7.0 | 43.6 | 61.3 | 55.0 | 67.7 | 41.5 |
| 20 | 22.5 | 12.9 | 46.8 | 66.0 | 58.9 | 71.0 | 46.4 |
| 30 | 30.4 | 18.7 | 51.6 | 70.0 | 63.4 | 75.6 | 51.6 |
| 40 | 33.6 | 20.9 | 53.0 | 73.8 | 68.8 | 77.0 | 54.5 |
| 50 | **37.8** | **28.5** | **53.7** | **74.2** | **69.6** | **77.6** | **56.9** |

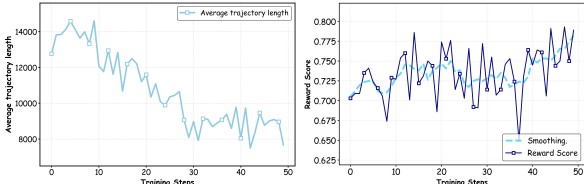

*Figure 3.* RL Curves of Mean Trajectory Length and Reward.

### 3.1. Main Results

Tab. 1 benchmarks proprietary and open MLLMs under three paradigms. Tool-free direct answering is consistently weak on open-domain multimodal deep-research tasks (*e.g.*, Qwen3-VL-30B-A3B-Thinking: 24.1% Avg.). ReAct-style agent workflows yield large improvements for most models (*e.g.*, Gemini-2.5 Pro: 50.7% Avg.), while naive RAG workflows provide limited gains on the reported settings.

Our Vision-DeepResearch models achieve the best performance among open models and are competitive with strong proprietary agentic systems. Under the same agentic backbone, Vision-DeepResearch-8B improves over Qwen3-VL-8B-Instruct (Agentic) by +10.4% Avg., with notable gains on MMSearch (+17.6%) and LiveVQA (+13.7%). Scaling to Vision-DeepResearch-30B-A3B further boosts results to 56.9 Avg. (+16.0), with consistent improvements on VDR (+17.6%), FVQA (+16.5%), and MMSearch-Plus (+18.5%), indicating that our data and training better instill long-horizon "reason-then-tool-call" behavior beyond generic agent prompting.

To further validate the effectiveness of our approach, we conduct an additional comparison with recent visual Retrieval-Augmented Generation (RAG) baselines (Wang et al., 2025a;b; Tanaka et al., 2025) on document-centric VQA benchmarks (Tanaka et al., 2023; Mathew et al., 2022; Ma et al., 2024). As shown in Tab. 5, our model consistently outperforms all existing methods by a significant margin. Notably, compared to the strongest baseline, ViDoRAG, our

to continue the rollout.

*2. Mask Trajectory*. We observe a severe negative-gradient issue during training. Concretely, for anomalous trajectories, for example, those terminated by the safeguards above, those exceeding the turn/length budget, or those dominated by format/tool-call errors (*e.g.*, accounting for more than half of the steps)—naively assigning a 0.0 reward is overly punitive. This can suppress otherwise correct steps within these trajectories, injecting substantial negative signal and ultimately destabilizing training. To address this, we mask such trajectories from gradient updates (*i.e.*, exclude them from backpropagation), while still including them in advantage computation.

*3. BF16 vs. FP16*. Recent study (Qi et al., 2025) suggests that RL training in FP16 can be more effective and stable than BF16. We also explored FP16. However, our rollouts are long (up to a 64K context), which led to numerical overflow and training instability under FP16. We therefore train in BF16.

In this section, we do not introduce algorithmic innovations, instead, these engineering techniques are crucial for ensuring stable large-scale agentic-reasoning RL training.

## 3. Experiments

In this section, we present a comprehensive experimental study. We first compare our approach with existing methods across a range of multimodal retrieval and reasoning benchmarks (Sec. 3.1). Then, we conduct ablation studies to analyze the contributions of key components, including both pipeline modules and data choices (Sec. 3.2 and Sec. 3.3). Finally, we analysis the RL training in Sec. 3.4. The experiment settings are presented in Appendix C.

approach achieves an absolute improvement of 14.1% on average, with a remarkable 20.4% performance leap on the highly challenging MMLongBench-Doc(MML-D) dataset.

### 3.2. Pipeline Ablation

We conduct a systematic ablation on 30B-A3B model to isolate the effects of multi-scale visual cropping and retrieval strategies (Tab. 2). We compare: Direct Answer (no retrieval), WIS (whole-image visual search), WIS+TS (whole-image visual + text search), CIS (multi-scale cropped-image visual search), CIS+TS (multi-scale cropping with both visual and text search), Image Caption+TS ( Image Caption + text search) and Image+TS ( Image + text search).

As shown in Tab. 2, the inability to use the image search tool leads to a significant performance degradation. This empirically demonstrates that visual evidence is inherently necessary. Also, removing retrieval (Direct Answer) yields very poor performance (Avg. 12.0%), confirming that external evidence is essential for open-domain multimodal reasoning. Whole-image visual search provides limited gains (Avg. 16.0%) and even degrades BC-VL (27.6%→26.1%), suggesting that single-shot retrieval is often distracted by background clutter and fails to surface long-tail knowledge. Adding text search on top of whole-image retrieval (WIS+TS) substantially improves overall accuracy (Avg. 29.3%), with a large boost on BC-VL (48.4%), highlighting the complementarity between visual grounding and textual evidence. Introducing multi-scale cropping for visual retrieval (CIS) dramatically improves VDR (4.8%→15.4%), validating the importance of localized object-centric anchors. However, without text search it remains suboptimal on knowledge-heavy benchmarks (MMS+ 22.7%; BC-VL 30.8%). The full pipeline (CIS+TS) achieves the best and most balanced results across all benchmarks (VDR 37.8%, MMS+ 28.5%, BC-VL 53.7%; Avg. 40.0%), indicating that multi-scale visual retrieval and text search are jointly necessary: cropping provides precise visual anchors, while text search supplies the missing long-tail factual evidence. The results in Tab. 3 also prove this insight.

As shown in Tab. 6, we additionally ablate the maximum inference turns and observe a continuous performance gain as the turn limit increases to 50, demonstrating that sustained multi-round interaction is essential for effective error correction and evidence collection.

### 3.3. Data Ablation

Tab. 4 shows that the base Qwen3-VL-30B-Instruct is insufficient for deep-research VQA (Avg. 24.3%; MMS+ 10.0%), indicating missing long-horizon tool-use and evidence grounding. SFT with tool-augmented trajectories brings major gains: adding verified VQA trajectories boosts MMS+ to 23.5% and BC-VL to 50.9%, while text-only QA

trajectories achieve similar improvements (MMS+ 23.5%; BC-VL 50.1%), validating effective transfer via our vision-to-text bridging pipeline. Adding fuzzy multi-hop VQA trajectories further improves MMS+ (26.0%) and BC-VL (51.4%), suggesting better coverage of long-tail, multi-hop settings. RL on top of SFT yields the best overall results (VDR 37.8%, MMS+ 28.5%, BC-VL 53.7%), showing that online interaction is crucial for refining long-horizon decision making beyond offline supervision.

### 3.4. RL Training

In Fig. 3, we visualize the RL training curves of average trajectory length and reward. The model initially tends to produce longer trajectories; as training proceeds, it learns to use the available tools more effectively, exhibiting shorter trajectories while achieving higher rewards. Consistent with the last two rows of Tab. 4 (post-SFT vs. post-RL), RL improves performance by an average of 2.4% on three challenging benchmarks, demonstrating the effectiveness of the large-scale RL training described in Sec. 2.3.2.

Moreover, due to API cost and wall-clock constraints, we do not exhaustively scale RL training. We expect the model to further benefit from larger-scale RL optimization.

## 4. Conclusion

We propose Vision-DeepResearch, which scales multimodal deep-research trajectories to dozens of reasoning steps and hundreds of search-engine interactions, yielding substantially stronger performance. We believe our work provides useful insights for the community.

## Impact Statement

Vision-DeepResearch advances multimodal deep research by enabling MLLMs to perform long-horizon, trial-and-error visual and textual retrieval in noisy real-world web environments. It scales reasoning depth and search breadth through multi-entity, multi-scale querying, and internalizes these behaviors via supervised fine-tuning and reinforcement learning—improving the robustness of multimodal agents in applications such as visual information seeking, product verification, scientific assistance, and accessibility tools, while reducing unsupported answers through stronger grounding in retrieved evidence.

## Acknowledgments

We acknowledge the support of GPU cluster built by MCC Lab of Information Science and Technology Institution, USTC. The AI-driven experiments, simulations and model training were performed on the robotic AI-Scientist platform of Chinese Academy of Sciences.

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

# A. Implementation Details.

To maintain consistency, we employ identical hyperparameters for both the 8B and 30B-A3B models across the Supervised Fine-Tuning (SFT) and Reinforcement Learning (RL) stages. Rather than conducting an exhaustive hyperparameter search, we adopt standard, widely used configurations. Specifically, during the SFT stage, we utilize a cosine decay learning rate schedule with a maximum learning rate of $2 \times 10^{-5}$ and a minimum learning rate of $5 \times 10^{-7}$. For the RL stage, we apply a constant learning rate of $1 \times 10^{-6}$.

In terms of computational cost, the SFT stage for the 8B model requires approximately 50 H800 GPU hours. For the 30B-A3B model, the SFT stage consumes roughly 110 H800 GPU hours, while the subsequent RL stage requires approximately 1,000 H800 GPU hours. There is no overlap between the RL and SFT data.

# B. Related Work

## B.1. Text-only DeepResarch LLMs

Early deep-research LLMs primarily focused on text-only environments. A series of works, including Tongyi-DeepResearch (Team et al., 2025), WebDancer (Wu et al., 2025b), WebSailor (Li et al., 2025), and WebShaper (Tao et al., 2025b), formulate complex information retrieval as an iterative loop of reasoning–tool call–re-reasoning, in which agents autonomously generate search queries, browse web pages, and iteratively integrate evidence to produce long chains of reasoning. This paradigm has led to substantial performance gains in open-domain question answering and knowledge-intensive reasoning tasks, demonstrating that "reasoning-then-tool-call" is an effective pathway toward more capable general intelligence.

However, these models rely almost exclusively on textual retrieval and access, where critical information is recalled through keyword-based text search. As a result, they lack essential capabilities such as visual perception, entity localization, and cross-modal consistency verification. Given that real-world information is inherently multimodal, text-based deep research agents struggle to support many high-value applications, including complex web page understanding, GUI comprehension, and product or visual entity recognition. Consequently, multimodal deep research agents capable of multi-round search, understanding, and decision-making in noisy visual environments are widely regarded as a key research direction toward more powerful artificial general intelligence (AGI).

## B.2. Multimodal DeepResearch MLLMs

Recent studies have begun to explore deep research capabilities in visual environments. WebWatcher (Geng et al., 2025) transforms text-based question answering into visual question answering via reverse image search, and constructs supervised fine-tuning datasets that enable agents to retrieve images and perform multi-step reasoning over them. MMSearch-R1 (Wu et al., 2025a) employs Group Relative Policy Optimization (GRPO) to incentivize models to actively invoke both image and text search tools, optimizing multimodal search strategies in an end-to-end manner. DeepMMSearch-R1 (Narayan et al., 2025) further introduces external grounding and cropping modules to isolate key regions in complex scenes before retrieval, partially mitigating background noise and improving retrieval effectiveness.

Despite these advances, existing multimodal deep-research MLLMs still suffer from critical limitations. (1) Coarse search strategy. Most prior work depends on one-shot full-level or entity-level image retrieval, making it difficult to reliably identify fine-grained visual entities through visual search engines in realistic web settings. The same for textual search, *i.e.*, reaching the desired target via a search engine often requires multiple attempts. (2) Insufficient optimization for long-horizon visual–text interaction. Current training paradigms typically focus on short-context or fewer-than-five-round retrieval scenarios, lacking systematic data and objective design for long-horizon (tens of rounds) visual and textual search. As a result, models often exhibit premature termination or shallow search behavior in complex tasks.

# C. Experiment Setups

**Training Data and Models.** Our training data and procedures follow the pipeline described in Sec. 2.2. We conduct both supervised fine-tuning (SFT) and reinforcement learning (RL) on Qwen3-VL-30B-A3B-Instruct (Bai et al., 2025), and only perform SFT on Qwen3-VL-8B-Instruct (Bai et al., 2025). For SFT, we use 30K high-quality visual deep research trajectories, while the detailed constitutions are presented in Sec. 2.3.2. Autoregressive supervision is applied at each step of every trajectory, covering the <think> reasoning, <tool_call> actions, and the final <answer>. This stage teaches

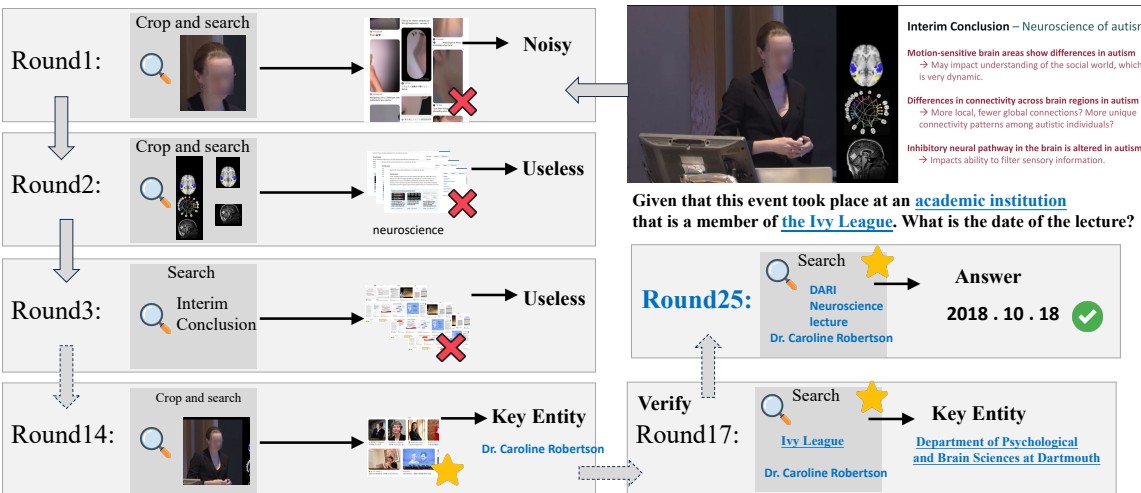

*Figure 4.* Our Data Pipeline. As shown in the top panel, we construct a complete multimodal deep-research synthesis pipeline. Leveraging the capabilities of an MLLM and a text-based DeepResearch foundation LLM, we generate long-horizon, multi-tool trajectories. As shown in the bottom panel, we obtain high-quality factual VQA instances via a rigorous verification and obfuscation procedure, which are then used for trajectory synthesis and RL training.

the model multi-round visual cropping, search, and reasoning behaviors. We adopt Ms-Swift as the training framework. For RL training, we use 15K high-quality VQA instances (described in Sec. 2.3.2) and sample complete trajectories through interaction with a real online search environment. An LLM-as-Judge is employed to evaluate answer correctness and provide reward signals, which are further combined with format constraints (*e.g.*, adherence to the ReAct template) for reward computation and policy optimization. This stage is implemented using the rllm framework (Tan et al., 2025).

**Benchmarks**. We evaluate our method on 6 challenging benchmarks, including VDR-Bench, FVQA (Wang et al., 2017), MMSearch-Plus (Tao et al., 2025a), MMSearch (Jiang et al.), LiveVQA (Fu et al., 2026), and BrowseComp-VL (BC-VL) (Geng et al., 2025).

Specifically, we use the test-mini split of VDR-Bench and the full split of BC-VL. We evaluate on all VQA instances in MMSearch and on the single-image subset of MMSearch-Plus. For FVQA and LiveVQA, we ***randomly select*** 300 samples per benchmark.

**Baselines.** We compare our method with a diverse set of proprietary and open-source multimodal models, covering both direct inference and agentic reasoning settings. The evaluated models include the Gemini-2.5 series (Comanici et al., 2025), Claude-Sonnet-3.7/4 models, GPT5 (Singh et al., 2025), the Qwen3-VL family (8B and 30B variants, Instruct and Thinking) (Bai et al., 2025), as well as recent open-source agents such as WebWatcher (Geng et al., 2025) and MMSearch-R1 (Wu et al., 2025a). These models are evaluated under two distinct reasoning paradigms. Direct answer refers to single-pass generation of the answer without calling any external tools or performing information retrieval. ReAct-style agentic reasoning follows a "reasoning-then-tool-call" paradigm, where the model iteratively performs reasoning, calls tools (including multi-scale image cropping, image search, text search, and web browsing) and integrates the retrieved evidence over multiple steps. To ensure a fair comparison, all models are equipped with a unified multimodal toolset and are evaluated using a consistent judge prompt for answer assessment.

## D. Case Study

As shown in Fig. 4,

