# OpenReview forum: "Vision-DeepResearch: Incentivizing DeepResearch Capability in Multimodal Large Language Models"
_ICML.cc/2026/Conference — ICML 2026 regular_

### Official Review · Reviewer_6JiS · 2026-02-23

**Soundness:** 3
**Presentation:** 2
**Significance:** 3
**Originality:** 2
**Overall Recommendation:** 4
**Confidence:** 4

**Summary:**

This paper proposes Vision-DeepResearch, which injects multi-round region-level image search and textual reasoning into MLLMs, giving them better capabilities to solve complex multimodal multi-hop problems requiring knowledge. To this end, Vision-DeepResearch designs a multimodal inference trajectory synthesis pipeline and performs engineering optimizations on RL training to improve training stability. Results on multiple downstream benchmarks validate the effectiveness of Vision-DeepResearch.

**Compliance With Llm Reviewing Policy:**

Affirmed.

**Final Justification:**

All my doubts are resolved, and I have decided to improve my score.

**Key Questions For Authors:**

Refer to the technical innovation, method design and description details, and experimental results in the weaknesses.

**Limitations:**

Yes

**Strengths And Weaknesses:**

Strengths:

1. Exploring multi-round, fine-grained retrieval and reasoning methods for complex multimodal multi-hop problems is meaningful.
2. Vision-DeepResearch shows significant improvements over baseline methods and achieves a level that can compete with large closed-source models with a small model size.

Weaknesses:

1. As the authors state, this paper does not offer any innovation in the SFT and RL training methods. While I acknowledge the importance of engineering techniques, this is, after all, an academic conference.
2. Why does a set $S_b$ with multiple bounding boxes only generate one action $A_t$? And is $A_t$ the same action at different times $t$? What is the purpose of labeling the subscript $t$?
3. For a complex multi-hop reasoning question, it's generally necessary to break it down into multiple sub-questions to answer step by step. Therefore, this may involve interleaved image and text search and reasoning. Vision-DeepResearch separates the image search and text search processes. Is this reasonable? Furthermore, can image search results alone accurately judge whether the evidence supports the answer?
4. What are the predefined high-quality image criteria in Factual VQA Verification? And this sentence, "Moreover, we submit the full image to an image search engine. If the retrieved results perfectly match the full-image query, we also discard the sample," I don't fully understand either. Isn't submitting the entire image to a search engine simply a full-image query?
5. In Fuzzy Multi-hop VQA Synthesis, how is a region cropped at multiple scales? What questions arise from alternating between answer obfuscation and entity obfuscation?
6. The Direct Answer metrics in Table 2 don't correspond to the results in Table 1, such as VDR's 4.8. In Table 3, the 8k text QA data in the SFT stage doesn't seem to offer much gain; what is the purpose of using it? Is there any overlap between the 15k samples in RL and the 30k in SFT?

---

> ### Author Rebuttal · Authors · 2026-03-31
>
> > W1: Contribution.
>
> We thank the reviewer and respectfully disagree with the characterization that our work is merely engineering. The training techniques are only one component; our main contributions are:
>
> 1. We identify an overlooked hit-rate problem in existing agentic workflows and DeepResearch MLLMs, motivating a framework for multi-turn, multi-entity, and multi-scale visual-textual search.
> 2. We show that current methods struggle on complex multimodal DeepResearch tasks due to limited reasoning horizon and restricted search interaction, and we formulate a long-horizon multimodal DeepResearch process.
> 3. We propose a new data pipeline for multimodal DeepResearch trajectories and multi-hop multimodal factual VQA data, which is missing in prior work.
> 4. We provide effective SFT and RL training strategies that are important for realizing the framework’s empirical gains.
>
> Thus, our contribution goes beyond implementation and includes problem formulation, system design, data construction, and an end-to-end framework. More broadly, impactful work does not necessarily require a new SFT or RL algorithm. Recent text-only DeepResearch works such as WebWalker (ACL 2025), WebDance (NeurIPS 2025), and WebWatcher (ICLR 2026) were published in academic conference without introducing new training algorithms. Their pipelines are arguably only a subset of our more comprehensive framework. We therefore kindly ask the reviewer to reconsider this point.
>
> > W2: Action $A_t$.
>
> The set $S$ corresponds to a single action $A_t$ because it is executed as one parallelized tool call containing multiple entity-level and multi-scale proposals. At reasoning step $t$, the model may issue multiple visual or textual queries within one function call, and we treat that call as one action. The subscript $t$ indexes the reasoning step, so different $t$ correspond to different actions across steps.
>
> > W3-1: Sequence of image search and text search.
>
> Prior multimodal DeepResearch frameworks, such as MMSearch-R1 and WebWatcher, also perform holistic image search before textual search, and our work follows and extends this setting. We believe this ordering is reasonable because identifying visual entities first and then retrieving textual evidence aligns with human reasoning.
>
> At the same time, our model can exhibit interleaved search and reflection in practice. As shown in Appendix Fig. 4, during textual search it may reflect on progress and return to visual search to gather additional evidence. We agree that more systematic study of interleaved search behaviors is an important future direction.
>
> > W3-2: Trajectory correctness.
>
> As detailed in our response to Reviewer YWvW Q2, we ensure synthesized trajectory quality using a visual evidence judge model together with strict rejection sampling. The effectiveness of this filtering is further supported by our experiments, where training on these trajectories yields clear performance gains.
>
> > W4-1: Factual VQA verification.
>
> Please refer to our response to Reviewer YWvW, W1-2.
>
> > W4-2: Full-image query.
>
> We exclude samples where a naive full-image search directly returns an exact match, since such cases are trivial and do not reflect real world scenarios. In real-world settings, querying a search engine with a noisy casual image (e.g., a photo of someone’s sneakers) is unlikely to produce an exact visual match. We therefore design the dataset to be more challenging and more representative of realistic multimodal DeepResearch tasks.
>
> > W5-1: Region crop.
>
> We prompt the MLLM to generate region proposals for distinct visual entities (e.g., different players in a sports scene), while also producing multi-scale crops for each entity, such as full-body, jersey-level, and facial views. This multi-entity, multi-scale strategy is important for improving robustness to search-engine hit-rate variation.
>
> > W5-2: Discussion for answer obfuscation and entity obfuscation.
>
> Please refer to our response to Reviewer YWvW Q3 for concrete examples of the interleaved answer and entity obfuscation process.
>
> A key concern is textual leakage, where the MLLM may exploit textual shortcuts instead of visual evidence. To prevent this, we apply strict verification and rejection sampling to discard VQA instances that can be solved using only parametric knowledge or textual hints. This ensures the dataset genuinely requires multimodal retrieval and reasoning.
>
> > W6-1: Error results in Tab. 1 and 2.
>
> Specifically, the Direct Answer results in Table 1 use Qwen3-VL-30B-A3B-Instruct, while Table 2 reports results for Vision-DeepResearch-30B-A3B.
>
> > W6-2: 8k text QA data.
>
> Although the 8k text QA data yields limited gains on MMSearch-Plus and BC-VL, where textual multi-hop reasoning is relatively simple and image matching is more critical, it substantially improves textual multi-hop reasoning and brings clear gains on VDR-Bench.
>
> > W6-3: RL data and SFT data.
>
> There is no overlap between the RL and SFT data.

---

> > ### Author Rebuttal · Reviewer_6JiS · 2026-04-05
> >
> > Thank you for the author's reply. My doubts have been resolved, and I have decided to raise my score to 4.

---

### Official Review · Reviewer_fXAW · 2026-03-11

**Soundness:** 2
**Presentation:** 3
**Significance:** 2
**Originality:** 2
**Overall Recommendation:** 5
**Confidence:** 4

**Summary:**

The authors introduce Vision-DeepResearch, a paradigm for multimodal deep research by training an MLLM (sft+RL) to perform multi-step visual and textual information retrieval and reasoning. As the authors highlight, instead of doing one-shot retrieval (like existing works) it mimics human behavior by iteratively refining the search (queries used) via cropping, adjusting scales, world-level modifications etc. Additionally for training  they construct long-horizon trajectories by combining visual search with text-based reasoning.

**Compliance With Llm Reviewing Policy:**

Affirmed.

**Final Justification:**

The author responses adressed my concerns raised during rebuttal.
They provided additional experiments supporting the claims made, the only matter i would like to see is performance on other tasks (apart from QA) but I understand this is too much for the rebuttal stage.

To sum up, I increased my score from 2 to 5.

**Key Questions For Authors:**

See Weaknesses + the following:

- It would be helpful if the authors clarify if they plan to release the training data, model checkpoints, and the full framework used.

- In the ablation study (e.g., Table 2), some benchmarks appear to be omitted. The authors should clarify why these datasets were excluded and whether the same trends hold when evaluating on the full set of benchmarks.

Less important points:

- Some engineering tricks occupy a relatively large portion of the paper despite not being central to the main contribution. These could be moved to the appendix to free space for more substantive analysis.
- Typo: Line 151 — “shwon” should be “shown.”

**Limitations:**

Mentioned in weaknesses

**Strengths And Weaknesses:**

Strengths:
- Overall the presentation is good. The paper is easy to follow and well written (apart from some engineering tricks/details that are not central the contribution and could be moved to Appendix)
- The idea of iteratively refining the query for the retrieval process is interesting and shows a degree of novelty.
- The authors include two useful ablations (on both the pipeline modules and the training data) that help clarify which parts contribute more to the overall performance.


Weaknesses (+suggestions):

- While the authors claim that an iterative refinement process of the query is necessary, the evaluation does not include a dedicated ablation analyzing this part. They do show improvements regarding the overall performance, but it would be helpful to show how performance evolves across retrieval calls or iterations, demonstrating whether and how the progressive refinement of queries improves the retrieved evidence (since it is one of the main contributions claimed)
- Limited comparison with closely related systems. I strongly believe the evaluation would be stronger if it included comparisons with closely related multimodal systems (that appear to be omitted). In particular, comparisons with approaches such as:
 MuRaR[1], M^2RAG[2], ViDoRAG[3], MuRAG[6], DeepMMSearch-R1, ColPali [5]. In this way the experimental will be more up-to-date.
- Narrow evaluation scope.  The work focuses exclusively on benchmarks for factual question answering. It would be informative to see whether the proposed paradigm generalizes to related multimodal retrieval tasks beyond factual QA. For example, benchmarks such as MMDocQA[3] or those used in ColPali [5] could enhance the evaluation scope. and enhance their evaluation suite with a border set of benchmarks.
- Incomplete related work discussion. The related work section appears somewhat limited and could be expanded to better reflect recent developments in the field. It would be nice to discuss more recent approaches like MuRaR[1], M^2RAG[2] and especially ViDo-RAG[3] which also explores iterative or agent-based retrieval workflows (see also the references below)

[1] MuRAR: A Simple and Effective Multimodal Retrieval and Answer Refinement Framework for Multimodal Question Answering (COLING 2025)

[2] Multi-modal Retrieval Augmented Multi-modal Generation: Datasets, Evaluation Metrics and Strong Baselines

[3] Vidorag: Visual document retrieval-augmented generation via dynamic iterative reasoning agents." (EMLNP 2025)

[4]MMDocRAG: Benchmarking Retrieval-Augmented Multimodal Generation for Document Question Answering (Neurips 2025)
[5] Colpali: Efficient document retrieval with vision language models (ICLR 2025)

[6] Murag: Multimodal retrieval-augmented generator for open question answering over images and text. EMNLP 2022.

---

> ### Author Rebuttal · Authors · 2026-03-31
>
> > W1: Ablation on iterative refinement.
>
> We maintain that iteratively optimizing the interaction between the model and the search engine through multi-round, multi-scale processes enables effective error correction and more comprehensive evidence collection, thereby yielding progressively more robust and accurate results. To further validate this claim, we conduct an additional ablation study by increasing the maximum number of turns from 1 to 50.
>
> | Max Turns | VDR | MMS+ | BC-VL | FVQA | MMSearch | LiveVQA | Avg. |
> | :--- | :---: | :---: | :---: | :---: | :---: | :---: | :---: |
> | 1 | 1.0 | 1.0 | 12.0 | 22.5 | 6.8 | 16.0 | 9.9 |
> | 10 | 14.2 | 7.0 | 43.6 | 61.3 | 55.0 | 67.7 | 41.5 |
> | 20 | 22.5 | 12.9 | 46.8 | 66.0 | 58.9 | 71.0 | 46.4 |
> | 30 | 30.4 | 18.7 | 51.6 | 70.0 | 63.4 | 75.6 | 51.6 |
> | 40 | 33.6 | 20.9 | 53.0 | 73.8 | 68.8 | 77.0 | 54.5 |
> | **50** | **37.8** | **28.5** | **53.7** | **74.2** | **69.6** | **77.6** | **56.9** |
>
> As shown in the table, our model demonstrates continuous performance improvement across all benchmarks with increasing inference rounds, particularly on more challenging benchmarks such as VDR-Bench and MMSearch-Plus.
>
> > W2&W3&W4: Qualitative comparison and discussion with RAG.
>
> We thank the reviewer for this suggestion. We will incorporate a detailed discussion of MuRaR, $M^{2}$RAG, and ViDo-RAG in the revised manuscript. Unlike traditional RAG models that rely on static corpora, Vision-DeepResearch operates in open-ended, noisy search environments using multi-turn, multi-step reasoning for iterative error correction, multi-hop evidence chaining, and long-horizon exploration.
>
> Regarding direct comparisons, each RAG model is tied to its own proprietary dataset. Despite our efforts to contact the authors to align evaluation settings, most were unable to release their private corpora due to proprietary restrictions. To address this, we utilized publicly available corpora to maximize alignment with standard evaluation settings and wrapped our Vision-DeepResearch agent tools following the VRAG evaluation protocols. We selected widely recognized benchmarks in the RAG community (SlideVQA, InfoVQA, and MMLongBench-Doc) and compared our approach against representative state-of-the-art methods in this field, including VDocRAG, ViDoRAG, and VRAG-RL. The results are shown in the table below.
>
> | Model| SlideVQA| InfoVQA |MMLongBench-Doc | **Avg.** |
> | :--- | :---: | :---: | :---: | :---: |
> | VDocRAG | 48.0 | 56.2 | 14.5 | 39.6 |
> | ViDoRAG | 82.5| 79.1 | 37.9 | 66.5 |
> | VRAG-RL | 56.2| - | 24.9 | - |
> | **Ours** | **93.2** | **90.4** | **58.3** | **80.6** |
>
> Even under zero-shot evaluation, Vision-DeepResearch significantly outperforms prior systems, demonstrating robust generalization to out-of-domain settings and noisy retrieval environments.
>
> Regarding DeepMMSearch-R1, since its model checkpoints are not publicly available, we directly compare our method with the results reported in their paper, as shown in the table below.
> | Model| InfoSeek|SimpleVQA | OKVQA | A-OKVQA |**Avg.** |
> | :--- | :---: | :---: | :---: | :---: |:---: |
> | DeepMMSearch-R1 | 47.5 | 55.9 | 67.8 | 73.5 |61.2 |
> | **Ours** | **49.0** | **72.0** | **80.3** | **82.6** |**71.0**|
>
> > Q1-1: Full open-source.
>
> We have already released our complete training framework codebase, model weights, and data examples to the research community. Furthermore, in line with our commitment, we will open-source the full dataset upon the acceptance of this paper to facilitate the complete reproducibility of our work.
>
> > Q1-2: Ablation study
>
> We chose VDR-Bench, MMSearch-Plus, and BC-VL for ablation experiments because they are more search-dependent and challenging compared to other benchmarks (such as FVQA, LiveVQA, and MMSearch). We supplemented the ablation validation with the remaining 3 benchmark sets and observed a trend consistent with previous ablation results, as shown in the table below.
>
> | Setting | FVQA | MMSearch | LiveVQA | **Avg.** |
> | :--- | :---: | :---: | :---: | :---: |
> | Direct Answer | 30.7 | 15.8 | 42.0 | 29.5 |
> | WIS | 44.3 | 27.5 | 47.3 | 39.7 |
> | WIS+TS | 67.2 | 61.4 | 71.3 | 66.6 |
> | CIS | 61.8 | 40.4 | 61.7 | 54.6 |
> | **CIS+TS** | **74.2** | **69.6** | **77.6** | **73.8** |
>
> | Setting | FVQA | MMSearch | LiveVQA | **Avg.** |
> | :--- | :---: | :---: | :---: | :---: |
> | Qwen3-VL-30B-Instruct | 57.7 | 55.0 | 60.0 | 57.6 |
> | +16K VQA traj. | 68.2 | 60.0 | 72.0 | 66.7 |
> | +8K QA traj. (SFT) | 70.3 | 62.4 | 74.3 | 69.0 |
> | +6K fuzzy VQA traj. (SFT) | 73.0 | 65.6 | 77.6 | 72.1 |
> | **+RL training** | **74.2** | **69.6** | **77.6** | **73.8** |
>
> > Q2: Less important points
>
> Thank you for your suggestion. We will revise the manuscript by moving the engineering tricks to the appendix, supplementing additional ablation studies, and correcting the typo on line 151.

---

> > ### Author Rebuttal · Reviewer_fXAW · 2026-04-03
> >
> > Thank you for the detailed response.
> > The authors have adequately addressed the questions that I raised by including additional experiments and my concerns are now resolved.
> > Accordingly, I will update my score.

---

### Official Review · Reviewer_YWvW · 2026-03-11

**Soundness:** 3
**Presentation:** 3
**Significance:** 3
**Originality:** 3
**Overall Recommendation:** 5
**Confidence:** 3

**Summary:**

This paper addresses the issue in Vision-DeepSearch, where related work often has limited reasoning depth and search breadth, making it difficult to solve complex questions that require aggregating evidence from diverse visual and textual sources. To solve this, the authors propose a new multimodal deep-research paradigm that performs multi-turn, multi-entity, and multiscale visual and textual search to robustly hit real-world search engines in the presence of heavy noise. The proposed method outperforms related works across 6 benchmarks.

**Compliance With Llm Reviewing Policy:**

Affirmed.

**Key Questions For Authors:**

1. This paper introduces a data generation pipeline and collects a training set. Will these be open-sourced to ensure the method is reproducible?
2. Quality issues from the previous data generation step could be passed into the later generation phases. How do the authors address this problem?
3. Questions generated during answer obfuscation, like "What is the name of the teacher of the cat owner’s daughter?", feel very unnatural. It's difficult to picture this being asked in real life. What is the author's perspective on this?

**Limitations:**

yes

**Strengths And Weaknesses:**

Strengths:
1. The vision deep search problem is very practical. The proposed method fills an important gap in the field.
2. The authors' method significantly outperforms previous approaches on 6 benchmarks.

Weaknesses:
1. The authors should clearly explain the data sources and quality control in the data pipeline section. For example, what are the open-source datasets mentioned in 'Factual VQA Verification'? And how exactly is the MLLM used as a selector?

2. There is a lack of experiments verifying the necessity of visual evidence. As a reader, I am not fully convinced that visual evidence is necessary, given the current ablations, since the method takes both the image and the text description. Without clarifying this point, the vision-deepsearch risks reducing to a combination of image understanding and text-based deep search. Therefore, the ablation studies should explore whether the visual evidence is truly needed (e.g., by dropping the image and using only the text description).

---

> ### Author Rebuttal · Authors · 2026-03-30
>
> > W1-1: Data sources.
>
> The open-source datasets utilized in our work comprise the training set of OVEN, LiveVQA, FVQA, and OKVQA.
>
> > W1-2: MLLM selector.
>
> The selection mechanism primarily focuses on two dimensions. The first is information richness: we prioritize images exhibiting complex visual features, specifically those containing multiple visual entities. In practice, this is implemented by employing an MLLM to predict bounding boxes for entities within the image, thereby favoring samples with a high density of entities.
>
> The second dimension is search necessity: we systematically filter out trivial, commonsense images (e.g., national flags) as well as poorly structured images (e.g., random doodles). Specifically, the former category encompasses samples where the MLLM can directly identify visual entities without relying on external search. For the latter, we explicitly prompt the MLLM to identify and discard samples where entities cannot be resolved due to poor structural quality.
>
> W2: Visual evidence necessity.
>
> | Setting | VDR | MMS+ | BC-VL | Avg. |
> | :--- | :---: | :---: | :---: | :---: |
> | Direct Answer | 4.8 | 3.6 | 27.6 | 12.0 |
> | Image Caption + TS | 9.0 | 16.7 | 42.0 | 22.6 |
> | Image + TS | 11.6 | 15.8 | 44.6 | 24.0 |
> | **CIS+TS (Ours)** | **37.8** | **28.5** | **53.7** | **40.0** |
>
> As shown in the table above, we have conducted two additional sets of experiments: "Image Caption + TS" and "Image + TS". These configurations represent providing the Vision-DeepResearch model with either an image caption or the raw image, respectively. In both settings, the image search tool is explicitly disabled, restricting the model to exclusively utilize the text search tool. As can be observed, the inability to use the image search tool leads to a significant performance degradation. This empirically demonstrates that visual evidence is inherently necessary.
> > Q1: Full open-source.
>
> We have already released our complete training framework codebase, model weights, and data examples to the research community. Furthermore, in line with our commitment, we will open-source the full dataset upon the acceptance of this paper to facilitate the complete reproducibility of our work.
>
> > Q2: Error propagation problem.
>
> To mitigate this issue, we employ multiple strategies across two main stages, i.e., VQA filtering/synthesis and trajectory generation, to meticulously ensure data quality.
>
> To reduce bias during the VQA construction process, we discard images smaller than 224$\times$224 pixels and leverage an MLLM-based selector to identify high-quality, real-world images containing multiple entities, while concurrently filtering out trivial samples. To further ensure the necessity of visual retrieval, we explicitly reject samples where the model can either answer the question without external evidence or achieve a perfect match via trivial whole-image search. During the base VQA synthesis phase, we design our pipeline to simulate the questioning process of human experts. Specifically, we prompt an MLLM to generate multiple candidate questions based on diverse retrieved evidence. Subsequently, a judge MLLM evaluates these candidates to select the most logically sound and objective version.
>
> To address potential quality degradation during trajectory construction, we employ a judge model during the visual trajectory generation phase. At each step, this judge evaluates the accumulated evidence and outputs a binary "hit" signal, ensuring that the search process gathers sufficient information for downstream reasoning. Following the completion of the textual trajectory, the model predicts the final answer. At this point, we apply a strict rejection sampling mechanism: an LLM verifies the consistency between the predicted final answer and the ground truth, discarding any trajectories that yield incorrect answers. This guarantees that all synthesized trajectories successfully resolve the corresponding queries.
>
> It is worth noting that while we have meticulously designed the aforementioned pipeline to minimize the error propagation problem, a marginal degree of error accumulation may still persist. We acknowledge this as an inherent limitation of current paradigms and consider it a promising direction for future work.
>
> > Q3: Answer obfuscation dicussion.
>
> The examples provided in the below are merely for reference purposes.
>
> "What brand is the football in the picture?" (root question)
>
> "What is the name of the founder of the football brand in the picture?" (Answer obfuscation)
>
> "What did the founder of the football brand in the picture do in Wilmslow, Cheshire, England?" (Answer obfuscation)
>
> "There is a town in Cheshire, England, located 11 miles south of Manchester. What is the famous thing the founder of the football brand in the picture did in this town?" (Entity obfuscation)
>
> When combining both answer obfuscation and entity obfuscation, the resulting become significantly more aligned with real-world question patterns.

---

> > ### Author Rebuttal · Reviewer_YWvW · 2026-04-03
> >
> > The response fully addressed my concerns.

---

### Official Review · Reviewer_PM1S · 2026-03-12

**Soundness:** 3
**Presentation:** 3
**Significance:** 3
**Originality:** 3
**Overall Recommendation:** 4
**Confidence:** 2

**Summary:**

The paper propose a novel multimodal deep-research framework targeting the problem that existing frameworks perform poorly on hit-rate and reasoning depth. The author propose to adopt a multi-scale search strategy that can significantly improve the hit-rate as the query is more atomic. To train the model to support such framework, the author propose a novel data pipeline and train the model. Experimental results shows the proposed method surpass existing methods significantly.

**Compliance With Llm Reviewing Policy:**

Affirmed.

**Final Justification:**

The rebuttal process solved my concerns and I keep my positive recommendation of 4.

**Key Questions For Authors:**

See strength and weakness.

**Limitations:**

yes

**Strengths And Weaknesses:**

Strength

1. The paper introduces multi‑entity, multi‑scale visual cropping and integrates it with text search, addressing the hit‑rate problem and allowing deeper searches.
2. The paper introduces a robust pipeline combines image verification, random walks, and obfuscation to generate complex VQA data.
3. Experiments are comprehensive with detailed ablation on the training data and other design choices.

Weakness

1. The framework significantly incurs a complex data construction and training and hyper-parameter set which may limits its real application. And the paper would benefit from a discussion on the computation cost.

---

> ### Author Rebuttal · Authors · 2026-03-30
>
> > W1-1: Complex of data construction, training pipeline and hyperparameters.
>
> We sincerely thank the reviewer for the constructive feedback.
>
> Regarding the concerns about the complex data construction and training pipeline: We acknowledge the complexity of our proposed framework, but we emphasize its necessity. As the agentic reasoning capabilities of foundation models become increasingly saturated, achieving further performance gains fundamentally grows more challenging. This inherently demands an increase in framework complexity; however, this complexity directly yields our SOTA performance. As demonstrated in Table 1 of the main paper, our base model, Qwen3-VL-30B-A3B, already surpasses the previous SOTA multimodal DeepResearch model, Webwatcher-32B, across multiple benchmarks. Despite this strong baseline, our final model still achieves an additional 16% improvement in average benchmark performance over the base model. This phenomenon is similarly observed in the text-only DeepResearch LLM domain; for example, recent works such as Tongyi-DeepResearch [1] and MiroThinker [2] also rely on highly intricate data synthesis pipelines, complex training strategies, and substantial computational overhead to enhance existing base models. To facilitate future research, we have comprehensively detailed our data synthesis and training methodologies (both our conceptual insights and engineering heuristics in the paper) to enable the community to effectively build upon our exploration to develop more powerful DeepResearch MLLMs.
>
> Regarding the hyperparameter settings: We utilize identical hyperparameters for both the 8B and 30B-A3B models across both the Supervised Fine-Tuning (SFT) and Reinforcement Learning (RL) stages. Furthermore, we did not perform exhaustive hyperparameter searching; instead, we adopted standard, widely used configurations. For instance, the SFT stage employs a maximum learning rate of 2e-5 and a minimum learning rate of 5e-7 with a cosine decay schedule, while the RL stage uses a constant learning rate of 1e-6.
>
> > W1-2: Computation Cost.
>
> Regarding the training overhead: We apologize for omitting the specific computational costs in the original manuscript. For the 8B model, the SFT stage consumes approximately 50 H800 GPU hours. For the 30B-A3B model, the SFT stage consumes roughly 110 H800 GPU hours, and the RL stage requires approximately 1000 H800 GPU hours.
>
>
> [1] Tongyi DeepResearch Technical Report.
>
> [2] MiroThinker: Pushing the Performance Boundaries of Open-Source Research Agents via Model, Context, and Interactive Scaling.

---

> > ### Author Rebuttal · Reviewer_PM1S · 2026-04-03
> >
> > The response solved my concerns.

---

### Decision · Program_Chairs · 2026-04-30

**Decision:**

Accept (regular)

**Comment:**

This paper focus on a key problem in the domain of visual deep search: multi-entity and multi-step retrieval. The authors enhance model capabilities in this area through a carefully designed data construction pipeline and reinforcement learning, and validate the effectiveness of their approach across multiple benchmarks. The topic is impactful and the claims are well supported. Furthermore, the authors have convincingly addressed all reviewer concerns during the rebuttal phase. Therefore, I recommend accepting this paper.